# Wildfire Spread Scenarios: Increasing Sample Diversity of Segmentation Diffusion Models with Training-Free Methods

Sebastian Gerard[*1] and Josephine Sullivan[1]

[1]KTH Royal Institute of Technology, Stockholm, Sweden
{sgerard, sullivan}@kth.se

## Abstract

Predicting future states in uncertain environments, such as wildfire spread, medical diagnosis, or autonomous driving, requires models that can consider multiple plausible outcomes. While diffusion models can effectively learn such multi-modal distributions, naively sampling from these models is computationally inefficient, potentially requiring hundreds of samples to find low-probability modes that may still be operationally relevant. In this work, we address the challenge of sample-efficient ambiguous segmentation by evaluating several training-free sampling methods that encourage diverse predictions. We adapt two techniques, particle guidance and SPELL, originally designed for the generation of diverse natural images, to discrete segmentation tasks, and additionally propose a simple clustering-based technique. We validate these approaches on the LIDC medical dataset, a modified version of the Cityscapes dataset, and MMFire, a new simulation-based wildfire spread dataset introduced in this paper. Compared to naive sampling, these approaches increase the HM IoU* metric by up to 7.5% on MMFire and 16.4% on Cityscapes, demonstrating that training-free methods can be used to efficiently increase the sample diversity of segmentation diffusion models with little cost to image quality and runtime.

Code and dataset: https://github.com/SebastianGer/wildfire-spread-scenarios

## 1 Introduction

Predicting wildfire spread is inherently uncertain, controlled by many interacting factors (fuel conditions, weather dynamics, topography), and often based on temporally sparse observations with limited spatial resolutions. The current literature [1–4] focuses on predicting the most likely outcome or an average of the possible futures. More effective disaster response can be enabled by anticipating *multiple* plausible futures instead, including low-probability scenarios that are still operationally relevant. Generative diffusion models [5–9] offer a principled way to learn and sample from such multi-modal outcome distributions, enabling the exploration of both

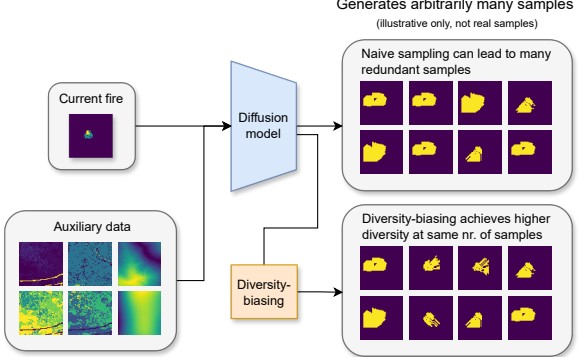

**Figure 1. Diversity-biased sampling:** We train a conditional diffusion model to generate different outputs for the same input data. If the goal is to find most, or all, different outputs for the current input, naive sampling can require a large number of samples, due to the redundancy in samples. To reduce this redundancy, we employ methods that bias the sampling towards higher diversity for the same number of samples.

common and rare wildfire spread scenarios given under-specified or uncertain conditions.

Building on this motivation, in this paper we examine whether diffusion models, trained in a supervised setting on segmentation masks that represent the variety of plausible outcomes, can be efficiently sampled at inference time to generate multiple distinct segmentation masks, that are consistent with the same inputs. This task, often termed *ambiguous segmentation*, arises not only in wildfire forecasting but also in medical imaging, where experts produce diverse segmentation masks for an input, or in autonomous driving, where multiple possible future scenarios must be considered. These domains share the challenge that the segmentation decisions are ambiguous and also that rare but valid segmentation masks must be efficiently identified.

Rather than introducing a new diffusion sampling setup, we focus on adapting and evaluating recent diversity-encouraging methods, originally developed for natural image generation, for ambiguous segmentation. We investigate particle guidance [10] and SPELL [11], two techniques that repel the samples in a batch to find distinct modes, studying how they transfer to the discrete segmentation outputs. We further show that SPELL's key diversity parameter

---

*Corresponding Author.

Proceedings of the 7th Northern Lights Deep Learning Conference (NLDL), PMLR 307, 2026.

can be directly related to dataset statistics, substantially reducing the need for hyperparameter sweeps. In parallel, we propose a simple clustering-based pruning strategy that reduces the number of fully denoised trajectories required, compared to naive sampling, when fidelity of the output masks outweighs the extreme computational efficiency where particle guidance and SPELL excel.

We evaluate these approaches across three domains with inherent ambiguity: the LIDC medical-imaging dataset [12], a Cityscapes-based ambiguous segmentation task [13–15], and MMFire, a wildfire spread benchmark we introduce that provides multiple simulated future outcomes per input. Our experiments assume multiple known outcomes for each input during training. We view this as a critical first step to the much harder but more realistic scenario for wildfire spread prediction, where only a single observed outcome is available per input.

For the datasets with more extreme variation between outcomes, the diffusion-based sampling methods produce significantly more diverse and distinctive outcomes than the de facto prior approach, the Probabilistic U-Net [14]. Moreover, particle guidance and SPELL consistently achieve superior quality-diversity trade-offs compared to naive diffusion sampling, while our clustering-based pruning yields further fidelity improvements without modifying the underlying sampling trajectories at a computational cost. Together, these results demonstrate that efficient diffusion sampling methods transfer effectively to ambiguous segmentation and offer a promising foundation for practical decision-support systems in domains where anticipating a spectrum of plausible futures is essential.

## 2 Related work

Research on generating segmentation masks with diffusion models either uses the *Gaussian diffusion* framework or variants of *categorical diffusion*. When generating binary segmentation masks [16–19], Gaussian diffusion can be used directly, followed by thresholding to binarize the real-valued outputs. To extend this from binary to multi-class segmentation masks, Analog Bits [20, 21] can be used with little change to the underlying mechanics.

In contrast to Gaussian diffusion, categorical diffusion [22–27] uses discrete state spaces, instead of real-valued ones. Empirically, both approaches perform similarly for segmentation [25, 28].

In this work, we use Gaussian diffusion. This allows us to integrate diversity-related methods [10, 11], that have been developed for Gaussian diffusion, more easily. We focus on binary masks, assuming that the results can be transferred to the multi-class setting via Analog Bits.

Most studies on diffusion segmentation models focus on achieving a high segmentation performance, by aggregating multiple samples as a form of implicit ensembling [17], or improving segmentation and calibration scores of the mean-aggregated samples [16]. However, these improvements could conceptually also be achieved with discriminative methods. We instead want to focus on the unique ability of generative methods to generate multiple *different* predictions for the same input. We are only aware of two studies [25, 29] that investigate the performance of their model on a dataset with *multiple* correct annotations, also termed *ambiguous segmentation*.

Various methods have been proposed in the diffusion model literature to increase sample diversity, usually focused on the text-conditioned generation of natural images. CADS [30] adds a noise schedule to the conditioning. This is supposed to prevent samples from focusing on the most probable modes, and instead explore more of the latent space. We found that CADS severely degrades the image quality and thus do not use it (see Appendix E for details).

Instead of modifying the conditioning to increase diversity, most methods modify the sampling schedule. Particle guidance [10] computes a guidance term based on the pairwise distances between noise-free predictions of the current in-batch samples to repel them from each other. Motion modes [31] extends this to include several additional guidance terms, that encourage properties in the generated data that particle guidance might otherwise not preserve. We directly use particle guidance, since our domain does not lend itself as easily to additional guidance terms.

ProCreate [32] aims to generate samples that differ from existing samples. For a more accurate distance computation, the method 'looks ahead' by denoising for several steps. It then computes a guidance term similar to particle guidance. We also investigate the case of generating multiple batches of data with repellence from previously-sampled images. In contrast to ProCreate, we only use a single-step denoising for distance computations, since we find that initial predictions are rather close to the final samples for binary segmentation masks.

Contrary to guidance-based methods, SPELL [11] does not indiscriminately repel all close samples from each other. Instead, if two samples lie within a pre-defined L2-distance of each other (the *shield radius*), SPELL repels them just enough to ensure that the distance is maintained. We use SPELL as an alternative to particle guidance.

## 3 Method

We use the EDM diffusion framework [33] to generate segmentation masks, conditioned on an input image. During training, we randomly select a single target and teach the model to generate it in a supervised manner. During inference, we generate

multiple masks by denoising multiple random noise samples with the trained diffusion model. Particle guidance[10] and SPELL [11] are used during the denoising process to increase the diversity among these generated masks. They work heuristically, by pushing the samples in a batch away from each other, thus increasing the diversity within a batch.

## 3.1 EDM diffusion framework

We follow the EDM framework [33] for our denoising diffusion models. The EDM model is based on the following ordinary differential equation (ODE):

$$\mathrm{d}\boldsymbol{x} = -t\nabla_{\boldsymbol{x}}\log p(\boldsymbol{x};t)\mathrm{d}t, \quad (1)$$

where $\boldsymbol{x}$ is a noisy mask (also called *latent*) and $t$ is the ODE time step. We use the default variance exploding formulation, where the standard deviation of the Gaussian noise is $\sigma(t) = t$. Thus, we also refer to $t$ as the *noise level*. The ODE is solved via numerical integration with a 2nd order Heun solver [33]. This integration starts from a Gaussian noise mask with a high noise level $\sigma = \sigma_{\max}$ and gradually denoises until a practically noise-free mask is reached with standard deviation $\sigma_{\min}$.

We train a denoising neural network $D_\theta$ to remove noise by minimizing the objective:

$$\mathbb{E}_{\boldsymbol{y}\sim p_{\mathrm{data}}}\mathbb{E}_{t\sim p_{\mathrm{train}}}\mathbb{E}_{\boldsymbol{\epsilon}\sim\mathcal{N}(\mathbf{0},t^2\mathbf{I})}\|D(\boldsymbol{y}+\boldsymbol{\epsilon};t) - \boldsymbol{y}\|_2^2, \quad (2)$$

where a segmentation mask $\boldsymbol{y}$ is sampled from the data distribution $p_{\mathrm{data}}$ ; the current time step $t$ is sampled from $p_{\mathrm{train}}$; and a noise image $\boldsymbol{\epsilon}$ is sampled from an isotropic Gaussian with standard deviation $t$. The forward diffusion process then simply consists of adding the noise $\boldsymbol{\epsilon}$ to the ground truth segmentation mask $\boldsymbol{y}$. We refer to the noisy segmentation mask as $\boldsymbol{y}_t$. By optimizing this objective, the denoising model $D_\theta$ learns to predict the noise-free $\boldsymbol{y}$, given $\boldsymbol{y}_t$ and the current time step $t$.

After training $D_\theta$, Equation 1 can be solved by approximating the score function:

$$\mathrm{score}(x,t) = \nabla_{\boldsymbol{x}}\log p(\boldsymbol{x};t) = (D_\theta(\boldsymbol{x};t) - \boldsymbol{x})/t^2. \quad (3)$$

To use the EDM framework for segmentation, the generated sample needs to be conditioned on an input image that we want to segment. Therefore, we sample pairs ($\boldsymbol{y}$, $\boldsymbol{c}$) from the data distribution with segmentation mask $\boldsymbol{y}$ and input image, or *conditioning*, $\boldsymbol{c}$. We pass $\boldsymbol{c}$ to the denoising network $D_\theta$ as an additional input. We implement this by concatenating $\boldsymbol{c}$ to the current noisy segmentation mask $\boldsymbol{y}_t$ in the channel dimension.

## 3.2 Increasing sample diversity

When generating natural images, the reverse diffusion process first determines low-frequency features

at high noise levels (e.g. *where* in the image we see a dog). As the sample moves towards lower noise levels, more high-frequency features are determined (e.g. the *details* of the face and then of the fur). However, in images that are binary segmentation masks, there are very few such high-frequency features, since all pixels take values of 0 or 1. This is highly relevant for any diversity-encouraging methods, since it means that the changes we care about are only possible near high noise levels.

Furthermore, in exploratory experiments, we found that the denoiser model's prediction $D_\theta(x_t, t_{\max})$ at the initial time step $t_{\max}$ is often relatively close to the final output already. This allows us to treat this first prediction as a proxy for the final sample. While this proxy is not perfect, it is a cost-efficient approximation that we employ.

### 3.2.1 Clustering-based sample-pruning

A straight-forward method to find all modes of a diffusion model's distribution is to simply generate a large number of samples. However, this will always incur a relatively high cost. Ideally, we would like to achieve this large-batch behavior, while keeping the cost low. To achieve this, we sample a large initial batch of pure noise, denoise it in a single step, and discard all samples that are deemed redundant. To decide which samples are redundant, we perform $k$-medians clustering, with $k$ equal to the number of modes that we expect. We discard all but the medians determined by the clustering and finish the reverse diffusion process for the corresponding samples. The benefit of this approach is that it only uses unmodified sampling steps, thus avoiding any negative impacts on image quality that modifications to the sampling trajectory could have. As a distance metric for clustering, we use the chamfer distance instead of L2 distance, since the former proved slightly better (see Table D.2).

### 3.2.2 Particle Guidance

A popular approach to increase the fidelity of generated natural images are guidance terms, like classifier guidance [7] or classifier-free guidance [34]. These modify the score function in Equation 1 by an additive term:

$$\mathrm{d}\boldsymbol{x} = -t\left(\nabla_{\boldsymbol{x}}\log p(\boldsymbol{x};t) + \alpha\,\nabla_{\boldsymbol{x}}\,g(\boldsymbol{x};t)\right)\mathrm{d}t, \quad (4)$$

where we call $g(\boldsymbol{x};t)$ the *guidance function* and $\alpha$ is a scalar that we call the *guidance strength*.

While classifier-free guidance increases fidelity, it *decreases* diversity [35]. Particle guidance (PG) [10] does the opposite, by improving the diversity among samples (also called *particles*) in a batch, possibly at the cost of image quality. The basic mechanic is to compute a gradient that increases the pixel-wise L2-distances between images, based on radial basis

function (RBF) kernels. This approach is purely heuristic: Samples are pushed apart from each other, but the directions in which they are pushed are not aligned with any information about the data.

Let $\{\boldsymbol{x}_i | 1 \leq i \leq B\}$ be a batch of $B$ noisy masks. To compute the value of the guidance function $g(\boldsymbol{x}_i; t)$, the denoising model first estimates the noise-free masks $\tilde{\boldsymbol{x}}_i$ in a single step for all $i$: $\tilde{\boldsymbol{x}}_i = D_\theta(\boldsymbol{x}_i; t)$. Next, the pairwise RBF-kernels $k$ between those noise-free masks are computed via

$$k\left(\tilde{\boldsymbol{x}}_i, \tilde{\boldsymbol{x}}_j; t\right) = \exp\left(-\frac{\|\tilde{\boldsymbol{x}}_i - \tilde{\boldsymbol{x}}_j\|_2^2}{h_t}\right), \quad (5)$$

with $h_t = m_t^2 / \log(B)$, where $m_t$ is the median value of $\|\tilde{\boldsymbol{x}}_i - \tilde{\boldsymbol{x}}_j\|_2^2$ within the current batch of masks.

The negative kernel sum aggregates all distance relationships from mask $i$ to all other masks:

$$g(\boldsymbol{x_i}; t) = -\sum_{j=1}^{B} k\left(\tilde{\boldsymbol{x}}_i, \tilde{\boldsymbol{x}}_j; t\right) \quad (6)$$

Finally, the gradient of this scalar sum is computed with regards to the noisy mask $\boldsymbol{x}_i$, backpropagating through $D_\theta$. This gradient is then used as guidance in Equation 4.

### 3.2.3   SPELL: SParse repELLency

In contrast to particle guidance, SPELL [11] only repels samples that are too close to each other. The original authors use the metaphor of a shield of radius $r$ around each sample. If a sample enters this protected area around another sample, it is pushed away to an L2 distance of $r$.

Furthermore, SPELL is not a guidance method. Instead of adding a term to the score function, it modifies the score function in Equation 3 by changing the noise-free prediction with an additive term $\Delta$. The modified score function for $\boldsymbol{x}_i$ becomes:

$$\text{score}_{\text{mod}}(x_i, t) = (D(\boldsymbol{x}_i; t) + \Delta_i - \boldsymbol{x}_i)/t^2, \quad (7)$$

with the additive term $\Delta_i$ computed as:

$$\Delta_i = \sum_{b, b \neq i} \sigma_{\text{relu}}\left(\frac{r}{\|\tilde{\boldsymbol{x}}_{0,i} - \tilde{\boldsymbol{x}}_{0,b}\|_2} - 1\right) \cdot (\tilde{\boldsymbol{x}}_{0,i} - \tilde{\boldsymbol{x}}_{0,b}) \quad (8)$$

This approach has the advantage of avoiding costly backward passes like in particle guidance. But since it changes the target of the denoising process at all sampling steps, there is a high risk for generating lower-quality samples. Therefore, we will limit SPELL to high-noise areas of the sampling process, to give the score function the chance to correct potential image quality issues caused by the repellence.

### 3.2.4   Inter-batch diversity

When generating *multiple* batches of samples, we want to encourage diversity across these batches. For this reason, we keep a memory bank containing the already-generated samples for the current input. For both particle guidance and SPELL, we add two more method variants: one that only repels samples in the current batch from the samples in the memory bank, and one that repels both from the memory bank and from samples in the current batch. This is also used in the original SPELL paper [11].

## 4   Multi-modal datasets

We use three different multi-modal datasets (i.e. having multiple targets per input). For MMFire and Cityscapes, we know all targets and explicitly set their probabilities. These datasets are thus very useful for evaluation, but the way their annotations are generated does not represent a real-life use case. LIDC offers a case of real-life ambiguous segmentation, where different annotations represent differing opinions between domain experts. The datasets also greatly differ in their inter-mode variances: MMFire's modes always overlap in the initial burned area with a medium amount of difference between modes. In Cityscapes, there are large-scale differences between modes, but also large-scale overlaps between some modes. In LIDC, annotations typically differ very little. These characteristics will become relevant when setting the hyperparameters for SPELL in subsection 5.4.

### 4.1   MMFire

We generated MMFire (MM = multi-modal) with the help of the Simfire [36] simulator. For a given geolocation, it downloads from the LANDFIRE program [37, 38] real-world data that is relevant for predicting how wildfires spread, namely: dead fuel moisture at extinction, fuel bed depth, oven-dry fuel load, surface to volume ratio, and elevation. Initial wind speed and wind direction are randomly generated. An initial fire is set near the center of the image. The well-known Rothermel equations [39] then deterministically spread the fire for a desired time, based on these initial conditions.

To generate a dataset with multiple different outcomes per initial condition, we first randomly pick a location in the western USA, where fuel is abundant and LANDFIRE provides data. We simulate a fire at that location for 10 minutes to have a non-trivial initial fire state. From this initial state, we branch into eight different futures by setting the wind direction to $i \times 45°, i \in \{0, 1, \ldots, 7\}$, constant across the simulation area of $64 \times 64$ pixels. For each wind direction, we then simulate another 10 minutes of

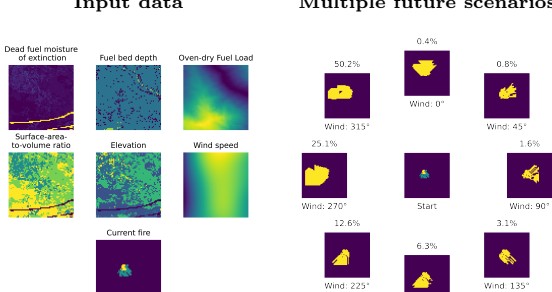

**Input data**

Dead fuel moisture of extinction • Fuel bed depth • Oven-dry Fuel Load

Surface-area-to-volume ratio • Elevation • Wind speed

Current fire

**Multiple future scenarios**

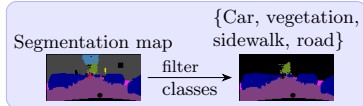

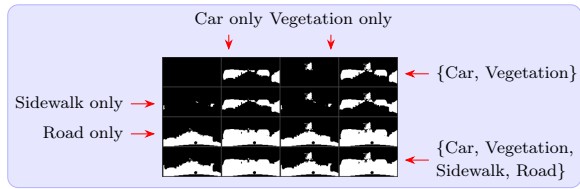

**Creating multiple targets for Cityscapes**

Segmentation map → filter classes → {Car, vegetation, sidewalk, road}

Independently flip classes to 0 or 1 with fixed probabilities, resulting in $2^4 = 16$ binary segmentation masks, or *modes*:

Car only • Vegetation only
Sidewalk only
Road only
{Car, Vegetation}
{Car, Vegetation, Sidewalk, Road}

**Figure 2. MMFire:** We use a wildfire spread simulator to generate multiple plausible outcomes based on the current state of the fire. This is done by setting the wind direction to one of eight values across the whole $64 \times 64$ image. We impose a highly skewed probability distribution on the eight outcomes during training (see the probabilities above). This represents a difficult situation where naively sampling from the diffusion model is a very slow strategy for finding all modes.

fire spread. We use the eight final states as the targets of the dataset. Figure 2 shows an example pair of input data and eight different futures.

Our models never get access to the wind direction. Instead, they are supposed to randomly pick a wind direction for each sample, according to the probabilities that we set. We impose a highly skewed distribution on the different modes (and their associated wind direction), weighting mode $i$ with a weight of $2^i$, to create a dataset in which it is challenging to find all modes of the distribution. With naive sampling, the expected number of samples to see each mode at least once is about 307.

## 4.2 Cityscapes: Multi-modal, binary version

Cityscapes [13] is a semantic segmentation benchmark dataset. Inspired by previous studies [14, 15], we synthetically make the annotations multi-modal. Unlike these previous studies, we stay within the binary regime, to stay closer to the data we are interested in, namely wildfire progression data.

To make the Cityscapes dataset multi-modal, previous studies split up one class into two new, synonymous, classes, e.g. *road* becomes $road_1$ and $road_2$. During training, we randomly choose whether all *road* pixels in the current image become $road_1$ or all become $road_2$. To stay binary, we instead flip those classes between the positive and the negative class. We do this with the classes *road*, *sidewalk*, *vegetation* and *car*. All other classes are always set to negative. We flip the classes to positive with respective probabilities 5%, 25%, 75%, 95%. Each class is individually flipped on or off, thus the combination of these flip decisions for all four classes leads to $2^4 = 16$ modes for each image, assuming

**Figure 3. Multi-modal binary Cityscapes:** The classes *road*, *sidewalk*, *vegetation*, and *car* are randomly flipped to the positive or negative class, with fixed probabilities, resulting in $2^4 = 16$ separate *modes* per image.

all four classes are present. This creates a skewed distribution, where naive sampling is a bad strategy to find all modes. Figure 3 shows all modes for an example image. While previous studies also used the class *person*, we generate segmentation masks at $64 \times 128$ pixels for faster experimentation. At this resolution, correctly annotating people is very difficult, so we drop this class.

## 4.3 LIDC

The LIDC dataset [12] contains CT scans of lungs and corresponding expert annotations of lung nodules. Each scan is annotated with four binary segmentation masks, that oftentimes disagree with each other. Crucially, if segmentation masks differ from each other, this represents actual disagreement between experts, stemming from epistemic uncertainty. The challenge when working with this dataset is that neither the full set of modes is not available.

## 5 Experiments

For more detailed information on implementation and experimental setup, please refer to Appendix B and Appendix C. For a visual comparison of the samples generated by the different methods, see Figure D.1 (MMFire) and Figure D.2 (Cityscapes).

### 5.1 Evaluation criteria

To evaluate generated samples, we compute the Hungarian-Matched IoU (HM IoU), finding the best match between generated and ground truth masks and computing the mean IoU across the matched masks. We modify this metric, indicated by HM IoU*, by de-duplicating the ground truth masks, since our goal is to *avoid* the generation of duplicates. For MMFire and Cityscapes, we know the

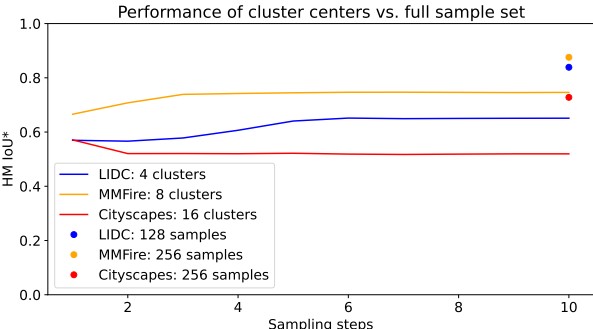

**Figure 4. Applying clustering-based sample-pruning at different sampling steps:** Varying after which sampling step the clustering and pruning is performed influences the final performance. For all datasets, there is large gap between evaluating only the cluster centers and evaluating the full set of generated samples. The x-axis represents the index of the sampling step. However, the noise levels at each step do not decrease linearly. See appendix for details on the noise schedule.

ground truth modes and can additionally compute an image quality metric and the number of distinct generated modes (see Appendix D). Results are averaged over five runs with different random seeds.

## 5.2 Clustering-based sample-pruning

Figure 4 shows the performance of clustering-based sample-pruning, applied at different sampling steps. For all datasets, we see a large difference between evaluating only the cluster centers (lines) and evaluating the whole set of generated samples (dots). This difference persists across all steps, even when the fully denoised samples are clustered at the last step. For Cityscapes, the difference between clustering at the final step and using the fully denoised batch of samples is 20.8% HM IoU*. This gap indicates that the clustering algorithm does not perfectly separate the available modes into different clusters.

We believe that this clustering failure is the result of two interacting factors: First, the modeling error: In approximating the conditional distributions over segmentation masks, the models produce imperfect outputs. Some of them are outliers and have a distort the cluster center choice. Second, the mode distribution asymmetry: We have purposefully chosen the MMFire and Cityscapes distributions over modes to be highly asymmetric, to create challenging benchmarks. Thus, rare modes will only be generated very seldom. In the context of outliers created by the imperfect models, it is then impossible for the clustering algorithm to decide which outliers are rare modes and which should be ignored.

We conclude from these results that clustering and pruning immediately after the first sampling step is the best option. Further steps increase the performance on MMFire and LIDC, but also incur

**Table 1. Clustering-based sample pruning: Varying initial batch size.** We investigate the trade-off between runtime and quality when varying the number of initial samples on Cityscapes. Naive sampling generates a number of samples equal to the batch size, while clustering always generates 16 samples, but starts with a higher number of samples that are clustered.

| Method | Batch size B | Image quality ↑ | Distinct modes ↑ | HM IoU* ↑ | Runtime ↓ |
|---|---|---|---|---|---|
| Naive sampling | 16 | 0.956 | 12.2 | 0.416 | 0h41m |
| | 32 | 0.956 | 13.3 | 0.497 | 1h19m |
| | 64 | 0.956 | 14.1 | 0.586 | 2h39m |
| Clustering [B→16] | 32 | 0.953 | 5.129 | 0.469 | 0h46m |
| | 64 | 0.951 | 5.680 | 0.517 | 0h53m |
| | 128 | 0.949 | 5.993 | 0.552 | 1h9m |
| | 256 | 0.948 | 6.122 | 0.571 | 1h47m |

the high cost of denoising all samples in the large batch again, while the performance gain is small.

Even with only one denoising step of the full batch, our method incurs a much higher computation cost than naive sampling, while generating the same amount of outputs. We reduce the number of samples in the initial large batch, to investigate the trade-off between performance and runtime. Runtime and performance drop step-wise with reduced batch size, as shown in Table 1. However our approach comes within 1.5% HM IoU* of naively sampling 64 samples, with only 16 generated samples. Depending on the use case, this superior sample-efficiency can be a very desirable property.

The strength of our clustering approach is not a low runtime, but a high sample quality. Thus, when comparing this method with others, we use 256 samples and disregard the high runtime. Table 2 shows that clustering outperforms the probabilistic UNet and naive sampling by at least 2.4% on MM-Fire, and 15.5% on Cityscapes. On Cityscapes it gets within 0.9% of the best performance among the investigated methods. On MMFire, it is clearly outperformed by both particle guidance and SPELL. We assume that this is caused by the two factors mentioned earlier. However, on LIDC, our approach matches the performance of the probabilistic UNet, outperforming the next-best diffusion-based method by 3.9%. This is likely because the rather uniform distribution of modes on LIDC makes it easier to determine correct cluster centers.

To compare with methods that generate *multiple* batches, we increase the number of clusters $k$ to the total number of desired samples, effectively over-clustering. Table 3 shows that clustering still beats naive sampling, but falls behind the other methods.

In summary, while the computational cost of this method is high, it can be useful if the goal is to produce a low number of representative samples, and the underlying distribution is not heavily skewed. Furthermore, it avoids potential image quality issues stemming from the interference with the sampling process, since it only follows the original sampling trajectories. Application cases for this method could be situations where a low number of high-quality

**Table 2. Single-batch performance:** We generate a batch of N samples per input; N is the number of modes of the respective dataset. Methods should produce a diverse set of samples while retaining a high image quality. PG: Particle Guidance. LIDC's ground truth data does not permit the computation of the image quality and distinct modes metrics.

| Method | Image quality ↑ | Distinct modes ↑ | HM IoU* ↑ | Runtime ↓ |
|---|---|---|---|---|
| MMFire - 1 batch × 8 samples | | | | |
| Naive sampling | 0.999 | 3.4 | 0.638 | 0h26m |
| Prob. UNet | 0.999 | 3.1 | 0.570 | 0h6m |
| Clustering [256→8] | 0.999 | 3.8 | 0.662 | 1h50m |
| PG: batch | 0.999 | 4.0 | 0.694 | 0h29m |
| SPELL: batch | 0.999 | 4.4 | **0.713** | 0h25m |
| Cityscapes - 1 batch × 16 samples | | | | |
| Naive sampling | 0.956 | 4.4 | 0.416 | 0h41m |
| Prob. UNet | 0.916 | 5.0 | 0.345 | 0h6m |
| Clustering [256→16] | 0.948 | 6.1 | 0.571 | 1h47m |
| PG: batch | 0.915 | 7.0 | **0.580** | 0h46m |
| SPELL: batch | 0.936 | 7.3 | 0.577 | 0h40m |
| LIDC - 1 batch × 4 samples | | | | |
| Naive sampling | | | 0.523 | 0h50m |
| Prob. UNet | | | 0.573 | 0h5m |
| Clustering [128→4] | n/a | n/a | **0.574** | 3h19m |
| PG: batch | | | 0.528 | 0h57m |
| SPELL: batch | | | 0.535 | 0h50m |

**Table 3. Multi-batch performance:** We generate two or four batches of N samples per input; N is the number of modes of the respective dataset. For clustering, we only generate one batch, but increase the number of clusters accordingly. PG: Particle Guidance. LIDC's ground truth data does not permit the computation of the image quality and distinct modes metrics.

| Method | Image quality ↑ | Distinct modes ↑ | HM IoU* ↑ | Runtime ↓ |
|---|---|---|---|---|
| MMFire - 2 batches ×8 samples | | | | |
| Naive sampling | 0.999 | 4.1 | 0.705 | 0h44m |
| ProbUNet | 0.999 | 3.8 | 0.637 | 0h12m |
| Clustering [256→16] | 0.999 | 4.4 | 0.728 | 2h8m |
| PG: batch | 0.999 | 4.8 | 0.751 | 1h4m |
| PG: memory bank | 0.999 | 4.6 | 0.738 | 1h4m |
| PG: batch & memory bank | 0.999 | 4.7 | 0.748 | 1h4m |
| SPELL: batch | 0.999 | 5.2 | 0.781 | 0h55m |
| SPELL: memory bank | 0.999 | 5.0 | 0.765 | 0h55m |
| SPELL: batch & memory bank | 0.999 | 5.3 | **0.784** | 0h55m |
| MMFire - 4 batches ×8 samples | | | | |
| Naive sampling | 0.999 | 4.7 | 0.751 | 1h28m |
| ProbUNet | 0.999 | 4.3 | 0.687 | 0h27m |
| Clustering [256→32] | 0.999 | 5.0 | 0.770 | 2h55m |
| PG: batch | 0.999 | 5.5 | 0.796 | 2h36m |
| PG: memory bank | 0.999 | 5.2 | 0.780 | 2h37m |
| PG: batch & memory bank | 0.999 | 5.3 | 0.786 | 2h37m |
| SPELL: batch | 0.999 | 6.0 | 0.826 | 2h17m |
| SPELL: memory bank | 0.999 | 5.5 | 0.801 | 2h19m |
| SPELL: batch & memory bank | 0.999 | 6.0 | **0.830** | 2h17m |
| Cityscapes - 2 batches ×16 samples | | | | |
| Naive sampling | 0.956 | 13.3 | 0.497 | 1h19m |
| ProbUNet | 0.915 | 6.0 | 0.427 | 0h12m |
| Clustering [256→32] | 0.948 | 7.3 | 0.661 | 2h23m |
| PG: batch | 0.916 | 8.5 | **0.696** | 1h37m |
| PG: memory bank | 0.936 | 7.7 | 0.647 | 1h36m |
| PG: batch & memory bank | 0.928 | 8.1 | 0.686 | 1h36m |
| SPELL: batch | 0.936 | 8.7 | 0.690 | 1h26m |
| SPELL: memory bank | 0.946 | 7.7 | 0.635 | 1h26m |
| SPELL: batch & memory bank | 0.936 | 8.7 | 0.690 | 1h26m |
| Cityscapes - 4 batches ×16 samples | | | | |
| Naive sampling | 0.956 | 14.1 | 0.586 | 2h39m |
| ProbUNet | 0.916 | 6.8 | 0.479 | 0h27m |
| Clustering [256→64] | 0.949 | 8.1 | 0.699 | 3h44m |
| PG: batch | 0.916 | 9.7 | **0.738** | 3h40m |
| PG: memory bank | 0.944 | 8.5 | 0.703 | 3h41m |
| PG: batch & memory bank | 0.938 | 8.9 | 0.723 | 3h41m |
| SPELL: batch | 0.936 | 9.7 | 0.735 | 3h19m |
| SPELL: memory bank | 0.951 | 8.3 | 0.680 | 3h19m |
| SPELL: batch & memory bank | 0.936 | 9.8 | 0.735 | 3h19m |
| LIDC - 2 batches ×4 samples | | | | |
| Naive sampling | | | 0.660 | 1h41m |
| ProbUNet | | | **0.715** | 0h7m |
| Clustering [128→8] | | | 0.695 | 4h1m |
| PG: batch | | | 0.677 | 1h54m |
| PG: memory bank | n/a | n/a | 0.675 | 1h54m |
| PG: batch & memory bank | | | 0.682 | 1h54m |
| SPELL: batch | | | 0.686 | 1h40m |
| SPELL: memory bank | | | 0.696 | 1h41m |
| SPELL: batch & memory bank | | | 0.697 | 1h41m |
| LIDC - 4 batches ×4 samples | | | | |
| Naive sampling | | | 0.727 | 3h32m |
| ProbUNet | | | **0.785** | 0h13m |
| Clustering [128→16] | | | 0.732 | 5h23m |
| PG: batch | | | 0.736 | 3h58m |
| PG: memory bank | n/a | n/a | 0.739 | 3h58m |
| PG: batch & memory bank | | | 0.743 | 3h58m |
| SPELL: batch | | | 0.740 | 3h30m |
| SPELL: memory bank | | | 0.751 | 3h31m |
| SPELL: batch & memory bank | | | 0.749 | 3h30m |

samples are presented to a human operator for analysis, but runtime is not a big concern.

### 5.3 Particle guidance

Following our intuition that the determination of modes for binary segmentation masks happens mostly in the early sampling steps, we limit particle guidance to the initial steps. Table D.1 shows that limiting the guidance to the initial step performs similarly well as computing it at every step, but saves computation by avoiding the backward steps necessary for computing the guidance term.

When choosing the strength of particle guidance, Table D.1 shows a trade-off between image quality and diversity. The loss in image quality is caused by the guidance pushing samples apart without any regard for the specific dataset or the learned score function. This can easily cause the samples to move into subspaces on which the denoising model has not been trained well, leading to faulty predictions.

### 5.4 SPELL

SPELL's main hyperparameter is the shield radius $r$. It defines an L2 distance within which no other sample is allowed to fall. If a sample violates this shield, it is pushed outside of the radius. In the case of binary segmentation masks, this L2 distance is equivalent to the square root of the number of pixels that must differ between two samples. This correspondence provides a very direct way to specify the desired diversity that does not exist for natural images (which use the full space in [0,1]) or for latent diffusion models (where distances in latent space do not directly correspond to distances in image space).

It becomes easy to set a shield radius, even if only single annotations are available in the training set.

We use the fact that we know several targets for each input to determine a starting value $r_0$ for the shield radius. For each input, we compute the minimum L2 distance among unique targets, and then compute the mean of these minima:

$$r_0 = \frac{1}{N} \sum_{i=1}^{N} \min \left( \{ \; ||y_{i,j} - y_{i,k}||_2 \; | y_{i,j} \neq y_{i,k} \} \right) \quad (9)$$

For each dataset, we conduct a coarse hyperparameter search around $r_0$ to determine the parameter to use. Table 4 shows this for Cityscapes. These results confirm that $r_0$ as computed above is a a

**Table 4. SPELL: Varying shield radius.** Based on diversity statistics on the Cityscapes training set, we perform a coarse search around $r_0 = 12.7$.

| Shield radius $r$ | Image quality ↑ | Distinct modes ↑ | HM IoU* ↑ |
|---|---|---|---|
| 6.350 | 0.944 | 6.6 | 0.540 |
| 9.525 | 0.934 | 7.4 | **0.564** |
| 12.700 | 0.924 | 7.8 | 0.561 |
| 15.875 | 0.914 | 8.1 | 0.551 |

**Table 5. SPELL: Limit application to high noise.** We vary $s_{\min}$, the highest noise level at which SPELL is still applied, to reduce the potentially negative influence during sampling. We begin sampling from pure Gaussian noise with $\sigma_{\max} = 80$, the default for the EDM framework. $s_{\min} = \infty$ represents never using SPELL. $s_{\min} = 0$ represents always using SPELL.

| $s_{\min}$ | Image quality ↑ | Distinct modes ↑ | HM IoU* ↑ | Runtime ↓ |
|---|---|---|---|---|
| $\infty$ | 0.956 | 12.2 | 0.416 | 41m |
| 70 | 0.938 | 7.0 | 0.571 | 41m |
| 40 | 0.936 | 7.3 | **0.577** | 40m |
| 20 | 0.935 | 7.4 | 0.571 | 41m |
| 10 | 0.935 | 7.4 | 0.571 | 41m |
| 0 | 0.934 | 7.4 | 0.564 | 0h40m |

very good initial value. For MMFire and LIDC, we found $r_0$ to be the best (see Table C.1).

The hard L2-limit enforced by SPELL means that samples are pushed apart without regard for how realistic the resulting images are. Especially towards the end of sampling, this is contrary to particle guidance, which slowly fades out with decreasing noise level. SPELL's stronger push seems to be unproblematic in the original SPELL paper [11], which uses latent diffusion models. These models have a downstream decoder model that maps the final latent representation to an image, which can potentially counter-act imperfect sampling outcomes. However, since we are directly applying the repellence in image space, we have to be more careful. To prevent SPELL from having a negative influence towards the end of sampling, we limit SPELL's application to $s_{\min} = 40$, where $s_{\min}$ is the highest noise level at which the guidance is still applied. In our case, that corresponds to the second sampling step. This change allows the score function to still guide the samples that were perturbed by SPELL towards more likely outcomes, leading to an improvement of 1.3% HM IoU* on Cityscapes (see Table 5).

**Cityscapes & MMFire:** On Cityscapes, particle guidance consistently beats all other methods, with a slim advantage of $\leq 0.6\%$ HM IoU* over SPELL. However, particle guidance's image quality is lower than that of SPELL by roughly 2%, and 4% lower than that of naive sampling, indicating a trade-off between quality and diversity. On MMFire, SPELL is clearly the best method, beating particle guidance by 1.9% HM IoU* on single batches and up to 3.4% in the multi-batch setting.

**Memory bank:** While particle guidance does not benefit from the addition of the memory bank, compared to simply using within-batch repellence in additional batches, SPELL achieves up to 0.4% improvement on MMFire when repelling both from the memory bank and the current batch items.

**LIDC:** Both particle guidance and SPELL clearly underperform the probabilistic UNet. This performance gap is reversed on the other datasets, where we assume that the highly skewed distributions are harder to model for the probabilistic UNet. However, the diversity-encouraging methods always outperform naive sampling from the diffusion model. Thus, when using diffusion models, it always appears advisable to use SPELL.

# 6 Future work

While MMFire is very useful for benchmarking, it lacks *realistic* diversity. Such diversity can be added to real-world datasets, which only have a single observed future for each input, by simulating alternative futures at each step. For our clustering-based approach, density-based clustering algorithms could make it easier to detect low-probability outliers as separate clusters. Furthermore, we only investigated training-free methods. Training-based approaches might improve upon these. Lastly, consistency models [40] might provide better one-step approximations, which all investigated methods rely upon.

# 7 Conclusion

In this paper, we investigated how to increase the sample diversity of diffusion models for ambiguous segmentation tasks, motivated by the application of wildfire spread prediction. The methods were evaluated across three datasets, including MMFire, an ambiguous segmentation dataset that we introduced in this paper. Our results demonstrate that the diversity-biased sampling consistently outperformed naive sampling, improving the HM IoU* metric by up to 16.4%. These findings provide a robust framework for generating distinct plausible outcomes in uncertain environments, paving the way for future work that transfers these results to observational real-world data.

# Acknowledgments

This work is funded by Digital Futures in the project EO-AI4GlobalChange. The computations were enabled by resources provided by the National Academic Infrastructure for Supercomputing in Sweden (NAISS) at C3SE partially funded by the Swedish Research Council through grant agreement no. 2022-06725.

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

# A  Datasets

## A.1  MMFire

The dataset consists of 9608 input samples of size $7 \times 64 \times 64$, each associated with eight simulated future fire spread segmentation masks of size $64 \times 64$. We use a split of 5000 training samples, 2500 validation samples, and 2108 test samples. We do not apply any augmentations. Any augmentations applied will need to make sure that the wind direction is correctly transformed, e.g. in rotations or flips.

## A.2  Cityscapes

Our implementation of the Cityscapes dataset is based on the existing Lightning Bolts [41] Cityscapes data module. We only use the 5000 images with fine annotations, resized to $64 \times 128$ for faster experiments. We keep the split of 2975 training images, 500 validation images, and 1525 test images. We use the semantic segmentation labels as a starting point, before filtering down to the four classes which we randomly flip between the positive and negative class. We only use color jittering as augmentations. We train with batch size 32 and parameterize the log-normal training noise distribution with a mean of $\mu_{\text{train}} = 1.5$.

## A.3  LIDC

To download and preprocess the LIDC dataset, we followed the steps indicated by the authors of [15] at https://github.com/gaozhitong/MoSE-AUSeg/, and use their dataset class to load the data. The dataset contains 9794 training images, 2314 validation images, and 2988 test images. It includes random horizontal and vertical flipping and rotation by up to 10°, all applied to both image and label simultaneously. The images and labels have a size of $128 \times 128$ pixels.

# B  Implementation details

Our code base is implemented in PyTorch [42] and PyTorch Lightning [43]. For the diffusion model, we use the official EDM [33] repository at https://github.com/NVlabs/edm.

All experiments were run on NVIDIA A40 GPUs, provided as part of a scientific computational cluster that is credited in the acknowledgments.

During sampling, the ODE is solved by starting from $\boldsymbol{x} \sim \mathcal{N}$ at $\sigma_{\max} = 80$ and numerically integrating to $\sigma_{\min} = 0.002$, using the 2nd order Heun solver from EDM. Unless specifically noted, we only use deterministic sampling, i.e. we set $S_{\text{churn}} = 0$.

We use the NCSN++ architecture [44], as implemented in the EDM code base and EDM preconditioning, modified to take an image as conditional information by concatenating it to the noisy mask $\boldsymbol{x}$ that is being generated. Only the base multiplier for the number of features is modified for the different datasets: For Cityscapes, we use the default value of 128, since the conditioning image and output distribution are rather complex. For MMFire and LIDC, we use 64.

# C   Experimental details

## C.1   Training the diffusion models

Unless mentioned otherwise, we train the base models with AdamW with learning rate 1e-4, and $\beta_1 = 0.9, \beta_2 = 0.99$. For Cityscapes, we train for 400 epochs, for LIDC, we train for 200 epochs, for MMFire, we train for 1000 epochs. We compute the validation loss after each training epoch and keep the model checkpoint with the lowest validation loss. During training, this validation loss is computed by randomly sampling noise levels according to the training noise distribution for each conditioning. The training noise distribution is always a log-normal distribution, with standard deviation $\sigma_{\text{train}} = 1.2$, which is the default value in EDM, and mean $\mu_{\text{train}}$, which we vary between different runs. Note that these parameters refer to the normal distribution, the samples of which are then exponentiated. The mean, mode, and standard deviation of samples drawn from the log-normal distribution are different.

For each dataset, we train several models, varying the mean ($\mu_{\text{train}}$) of the log-normal distribution used for sampling the noise levels during training. In preliminary experiments, we observed that performing model selection simply via lowest validation loss did not lead to a good calibration of the distribution over modes. We therefore perform the model selection with regards to the highest alignment of the sample distribution over modes with the training distribution. For this, we sample 64 segmentation masks per conditioning (e.g. per RGB image in Cityscapes), and compare the distribution over modes with the known ground truth distribution via total variation difference metric (see next paragraph). We also do not perform model selection with regards to HM IoU*. Choosing a model with the highest HM IoU* for a highly skewed distribution would mean that the skewedness is likely not properly represented by the model, even though we of course want to achieve a high HM IoU* in the end. In practice, the models we choose tend to still be among the best in terms of HM IoU* computed over the 64 samples. For datasets for which we only know a set of segmentation masks per image, but not the actual probabilities per mask, we can take the pixel-wise mean across generated masks and compare to the pixel-wise mean across ground truth labels, as a measure of calibration. This is the case for LIDC, where we use the Brier score to select a model. We end up choosing the models trained with the following parameters: MMFire: $\mu_{\text{train}} = 0.5$, Cityscapes: $\mu_{\text{train}} = 1.5$; LIDC: $\mu_{\text{train}} = 1.0$.

To perform **model selection**, we estimate how well a large batch of generated segmentation masks $\{x'_i | 0 \leq i \leq B \; B \in \mathbb{N}\}$ follows the training distribution over modes. For **Cityscapes**, we have flipped all classes separately, thus we want to estimate how well the model follows the per-class Bernoulli flip probabilities. We estimate the flip probabilities from the $B$ generated masks for each class separately. For this, we compute the per-class IoU between generated mask and indicator mask of the respective class, e.g. a mask that is 1 for the road class, and 0 otherwise. Then, we threshold the IoU at 0.5 to decide whether the generated mask represents a choice of the positive or negative mode for the given class. From these per-mask modes for each class, we estimate the per-batch distribution of modes and compare it to the true Bernoulli distributions via mean total variation distance (TVD). To compute this distance, we compute the mean distance between the $C$ paired distributions: $\text{TVD}_{\text{mean}}(\{(p_c, q_c) | 1 \leq c \leq C)\}) = \frac{1}{C} \sum_{c=1}^{C} |p_c - q_c|$, where $p_c, q_c$ are the true and estimated Bernoulli probability for flipping class c to the positive class. For MMFire, we do conceptually the same, except that we assign a single mode to each image, and we compare a single empirical and a single ground truth *categorical* distribution, instead of several Bernoulli distributions.

## C.2   Sampling from the diffusion models

For all models, we use the EDM sampling schedule, parameterized by $\rho = 7$ and $n = 10$ time steps. More steps did not lead to better results. This is likely due to the very conditioning signal that is much stronger than in the case of natural images, conditioned on a text prompt, for example. There are many different images that are consistent with 'dog on the beach wearing sunglasses', but very few segmentation masks that are pixel-perfectly aligned with the input and correctly distinguish between the positive and negative classes.

## C.3   Particle guidance and SPELL

For particle guidance, we use the a guidance strength $\alpha = 10$ for LIDC and $\alpha = 25$ for the other datasets, following separate grid searches over $\alpha \in \{2.5, 5, 10, 25, 50, 100, 1000\}$.

For SPELL, we compute $r_0$ from the respective training dataset, according to Equation 9. Starting from this, we run a coarse grid search across $\{0.5r_0, 0.75r_0, r_0, 1.25r_0\}$ and choose the best one as $r$ for all experiments. Table C.1 shows the corresponding values. For both MMFire and LIDC, using $r_0$ proved best among the investigated values. For Cityscapes, we used $0.75r_0$. Which exact value proves best depends on the exact distribution of distances. An option would be to replace the mean with the minimum in Equation 9. However, for many cases, SPELL would then not ensure enough diversity (see Table 4). Thus, $r_0$ can be taken as

**Table C.1. SPELL: Shield radii used in experiments.** $r_0$ is an initial estimate for a good shield radius determined from dataset diversity statistics in Equation 9. $r$ is the best value we found in a coarse grid search around $r_0$. For MMFire and LIDC, $r_0$ was the best value we found. For Cityscapes, we found in Table 4 that $0.75 r_0$ performs slightly better.

| Dataset | $r_0$ | r |
|---|---|---|
| MMFire | 9.525 | 9.525 |
| Cityscapes | 12.700 | 9.525 |
| LIDC | 6.000 | 6.000 |

a strong starting point, but, depending on the concrete distance distribution of the dataset, a better value might exist near $r_0$.

## C.4 Probabilistic UNet

Following previous work [25, 29], we use the probabilistic UNet [14] as a baseline for ambiguous segmentation. To stay within the PyTorch framework, we use an unofficial PyTorch re-implementation[1] to train one model per dataset. In our experiments, we try to stay as close as possible to the original hyperparameter choices that were made for training on LIDC in the original probabilistic UNet paper.

For Cityscapes and MMFire, training proved very unstable at first. We identified a logarithm operation in the log probability computation for the prior and posterior net, which could be stabilized by adding $\epsilon = 10^{-6}$ to the standard deviation of the respective normal distribution. The LIDC learning rate scheduler was set to reduce the learning rate at fixed numbers of epochs, for which it is unclear how this should be transferred to other datasets. Instead of manually setting such fixed points, we use the OneCycleLR[45] learning rate scheduler, which smoothly increases and decreases the learning rate, dependent on the total number of steps. We also increased the learning rate from 1e-4 to 1e-3 since convergence was infeasibly slow otherwise. We trained for $10^6$ steps and kept the model state with the best validation loss.

## D Evaluation

Metrics for the main results in Table 2 and Table 3 are computed on the test sets. All other tables and figures are treated as part of hyperparameter search or optimization, therefore they are computed on the validation sets. For Cityscapes, the test set segmentation masks are not public, therefore we can not compute any of our metrics on the Cityscapes test set. Therefore, we use the Cityscapes validation

---

[1] https://github.com/stefanknegt/Probabilistic-Unet-Pytorch

**Table D.1. Particle guidance: Varying guidance strength $\alpha$.** We investigate the trade-off between image quality and diversity on Cityscapes. Applying guidance on the first step only saves several backward passes, but reaches almost exactly the same performance.

| Guidance steps | $\alpha$ | Image quality ↑ | Distinct modes ↑ | HM IoU* ↑ |
|---|---|---|---|---|
| None | 0 | 0.956 | 12.2 | 0.416 |
| First | 1 | 0.953 | 5.2 | 0.477 |
| First | 2.5 | 0.950 | 5.8 | 0.518 |
| First | 5 | 0.946 | 6.1 | 0.550 |
| First | 10 | 0.938 | 6.5 | 0.575 |
| First | 25 | 0.915 | 7.0 | **0.596** |
| First | 50 | 0.881 | 7.6 | 0.595 |
| First | 100 | 0.827 | 8.3 | 0.568 |
| First | 1000 | 0.695 | 8.4 | 0.455 |
| all | 1 | 0.953 | 5.3 | 0.476 |
| all | 2.5 | 0.950 | 5.8 | 0.521 |
| all | 5 | 0.946 | 6.1 | 0.551 |
| all | 10 | 0.877 | 7.6 | 0.593 |
| all | 25 | 0.914 | 7.0 | **0.595** |
| all | 50 | 0.877 | 7.6 | 0.593 |
| all | 100 | 0.821 | 8.3 | 0.566 |
| all | 1000 | 0.689 | 8.5 | 0.455 |

set everywhere, instead. Since we mostly care about the relative performance of the methods on the same dataset, this still seems serviceable.

While we mainly focus our evaluations on HM IoU*, which is a combined measure of image quality and diversity, we additionally use explicit measures of these two qualities on MMFire and Cityscapes. For each generated sample, we compute which ground truth mode is closest to the sample. Within a batch of samples, we then count the number of **distinct modes** (or unique modes) that were generated. Since this is a pure argmax computation, a sample that is closest to one particular mode might still have very low quality. Thus, this metric can be very noisy and should always be interpreted with regards to an image quality metric.

For **image quality**, we compute a pixel-wise union of all modes for the current input, to determine which pixels are allowed to be part of the positive class, and which ones should always be assigned the negative class. We want to penalize samples which set pixels to positive that should never be positive. Let this union image be $y_{\text{union}}$. We take the complement to receive $\bar{y}_{union}$, which is 0 in all pixels that are positive in at least one mode, and 1 otherwise. Let $x_i$ be a sample to evaluate, then we compute the image quality metric as $1 - \text{IoU}(x_i, y_{\text{union}})$.

The runtimes we measure include all of the time it takes to run the respective job on a shared scientific computation cluster, including loading the respective model and computing the metrics. The runtimes can vary, depending on the load imposed by other jobs using the shared resources. We assume that they are still useful as broad indications of how much longer certain methods take than others.

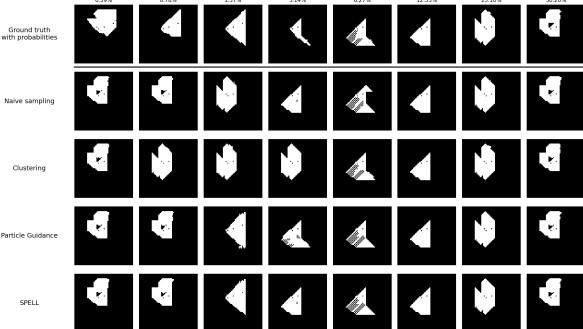

**Figure D.1. Method comparison - MMFire.** We compare the different diversity-encouraging methods on the MMFire dataset, sampled from the same starting noise and conditioning. The order of generated samples is determined via Hungarian matching, such that the samples are positioned below the closest ground truth. This example is cherry-picked for visualization. Non-cherry-picked examples often show the same number of correct samples across most methods, or can have duplicate ground truths. The low difference in visual appearance makes sense when we consider that the best method, SPELL, only performs 7.5% better than naive sampling, and that all methods use the same underlying diffusion model.

**Table D.2. Clustering - comparing distance functions:** We use our clustering-based approach to prune from 256 initial samples to 16 samples on Cityscapes, using either chamfer or L2 distance. L2 distance performs slightly worse, so we use chamfer distance for all experiments.

| Distance function | Image quality ↑ | Distinct modes ↑ | HM IoU* ↑ | Runtime ↓ |
|---|---|---|---|---|
| Chamfer | 0.948 | 6.1 | 0.571 | 1h47m |
| L2 | 0.952 | 5.9 | 0.560 | 1h33m |

# E  CADS

By adding a noise schedule to the conditioning information, CADS [30] aims to prevent the denoising process from immediately moving towards high-probability modes, thereby increasing diversity. We implement this by simply using the same noise schedule for the conditioning as for $x_t$. We then re-normalize the noisy conditioning to have an expected standard deviation of 1, which is the same as the noise-free Cityscapes data. Without this, the denoising model would have no chance to work well, since the noisy values would be out of distribution, compared to the training data. Since segmentation very directly relies on the conditioning, we also attenuate the noise level on the conditioning with a factor $\gamma \leq 1$. This should allow the model to access noisy conditioning information when it has to use this information for denoising. Given a conditioning image $c$, we therefore compute the noisy image $\hat{c}$ as:

$$\hat{c} = \frac{c + \gamma\epsilon}{1 + t^2}, \epsilon \sim \mathcal{N}\left(0, t^2\mathbf{I}\right) \qquad (10)$$

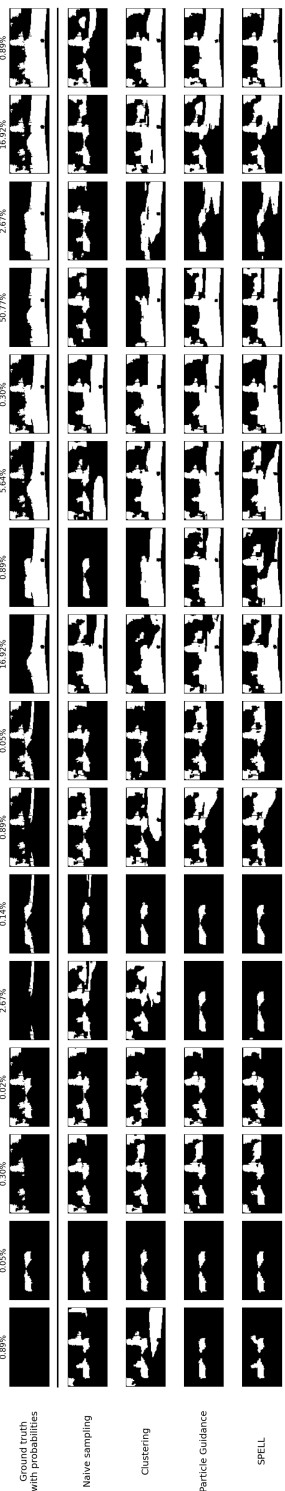

**Figure D.2. Method comparison - Cityscapes.** We compare the different diversity-encouraging methods in the Cityscapes dataset, sampled from the same starting noise and conditioning. The order of generated samples is determined via Hungarian matching, such that the samples are positioned below the closest ground truth. This example is cherry-picked for visualization to ensure that the different ground truth modes are easily enough visually distinguishable.

| Attenuation $\gamma$ | Image quality ↑ | Distinct modes ↑ | HM IoU* ↑ |
|---|---|---|---|
| 0.00 | 0.956 | 4.4 | 0.416 |
| 0.10 | 0.983 | 1.5 | 0.142 |
| 0.25 | 0.995 | 1.1 | 0.101 |
| 0.50 | 0.999 | 1.0 | 0.085 |
| 1.00 | 1.000 | 1.0 | 0.078 |

**Table E.1. CADS:** We impose a noise schedule on the conditioning information, attenuated by a factor $\gamma$. We see that any noise added to the conditioning information greatly hurts performance.

CADS works well in the original publication for text prompt embeddings, which are inherently continuously-valued, such that noisy embeddings are likely still within in-distribution. For image-conditioned generation, the diffusion models were never trained with a noisy conditioning, meaning that they are likely unable to extract information from it well. Furthermore, the denoising model mostly needs to generate low-frequency information, since the segmentation masks that the model generates only consist of binary values, without any fine-grained differences between them. However, such values are settled on *early* in the reverse diffusion process. At that point, CADS is obscuring most of the conditioning information, making it very difficult to generate high-quality images. This is very different for text-based image generation, which is much less reliant on correct low-frequency information. Table E.1 shows the corresponding experimental results: HM IoU* greatly suffers, even when only 10% of the original noise schedule is used.

