# OpenReview forum: "Wildfire Spread Scenarios: Increasing Sample Diversity of Segmentation Diffusion Models with Training-Free Methods"
_NLDL.org/2026/Conference — NLDL 2026 Poster_

### Official Review · Reviewer_D1xz · 2025-10-07
**Accept**

**Rating:** 4
**Confidence:** 3
**Final Rating:** 4
**Final Confidence:** 3

**Summary:**

The paper proposes a training-free diversity-enhancing sampling method for diffusion-based segmentation models. It integrates Particle Guidance and SPELL—two repulsion-based strategies—into the diffusion sampling process to generate more diverse outputs for the same input. A clustering-based pruning algorithm and a new wildfire simulation dataset, MMFire, are also introduced. Experiments on MMFire, Cityscapes, and LIDC show that the methods significantly outperform naive sampling. The approach is theoretically sound and well validated.

**Strengths:**

- The diversity sampling framework combining Particle Guidance and SPELL is theoretically consistent with diffusion models and clearly implemented, modifying only the sampling process.
- The approach enhances diversity without retraining, making it efficient and suitable for resource-limited or frozen-model scenarios.
- Experiments on MMFire, Cityscapes, and LIDC demonstrate the method’s robustness and strong potential for broader applications.
- The use of HM IoU* effectively balances quality and diversity, with comprehensive analyses across single, multi-, and inter-batch settings.
- The paper is well structured and easy to follow, with precise formulas, clear figures, and thorough appendices.

**Weaknesses:**

The diffusion backbone is domain-specific (segmentation-focused EDM) and small-scale. It remains uncertain whether the proposed diversity-biased sampling can transfer to general latent diffusion models such as Stable Diffusion or Flux, which would better demonstrate its scalability.

**Final Justification:**

Thanks for the rebuttal. I’ll keep my score unchanged.

**Justification:**

While the overall novelty is moderate as the paper is basically combining two training-free sampling techniques and the approach is tested only on small, domain-specific diffusion models, the results are convincing and of practical relevance. The study provides valuable insights into training-free diversity control in diffusion models, and could potentially be used for more general tasks. Thus, I recommend acceptance.

---

> ### Author Rebuttal · Authors · 2025-10-22
>
> Thank you for investing your time to assess our paper. We are glad that you find the results convincing and relevant. Below, we address your concern about transferring our results to latent diffusion models.
>
> **Transfer to latent diffusion models**
>
> Particle Guidance and SPELL were applied to Stable Diffusion in their respective original publications and work well to increase the sample diversity among natural images in this setting. Since we need much fewer denoising steps to generate high-quality segmentation masks than natural images, the performance gains from moving to latent diffusion models would be much smaller for our case than for the generation of natural images. Furthermore, we assumed that it would be easier to keep the precise pixel-wise correspondence between input images and generated segmentation masks intact, which is crucial for segmentation tasks, if we avoid the transition into and out of latent space. For these reasons we decided that it would be most sensible to only use pixel-space models in this work.

---

### Official Review · Reviewer_N3hm · 2025-10-07
**Relevant evaluation of training-free diversity techniques in diffusion-based segmentation methods and dataset contribution for wildfire spread prediction**

**Rating:** 4
**Confidence:** 4

**Summary:**

The paper introduces MMFire, a simulated dataset for evaluating wildfire spread prediction under uncertainty. Using a conditional diffusion model trained on segmentation masks, the authors study existing training-free methods—Particle Guidance, SPELL— and their own clustering-based pruning approach to enhance sample diversity without retraining. Experiments on MMFire, Cityscapes, and LIDC show that these methods increase diversity with minimal quality loss, with SPELL achieving the best trade-off.

**Strengths:**

- **Valuable dataset contribution**: the paper introduces MMFire, a simulated dataset designed for controlled evaluation of wildfire spread prediction under uncertainty. This is a meaningful and timely addition to the community, enabling reproducible studies on multi-modal prediction and uncertainty quantification in an application domain of clear societal relevance.

- **Sufficient technical contribution:** the methodological aspect is not groundbreaking but remains **adequate and relevant**. The authors adapt several **existing training-free diversity-inducing techniques**—Particle Guidance, SPELL, and propose a clustering-based pruning approach—to segmentation diffusion models. These adaptations are competently executed and yield useful empirical insights into how diversity can be encouraged without retraining.

- **Significant empirical evaluation:** the paper includes consistent experiments across three datasets (**MMFire**, **Cityscapes**, and **LIDC**), providing a convincing empirical evaluation of the proposed methods. The results show that diversity can be improved substantially with only minor quality degradation, and the ablations (e.g., on SPELL’s shield radius) are informative.

- **Presentation:** the paper is well structured and readable, with a clear description of the diffusion setup and experimental procedures, and only a few suggestions for better flow in the introduction. The commitment to releasing both the **dataset and code** further strengthens its practical impact.

**Weaknesses:**

- **Introduction motivation and flow:** the introduction jumps quickly from uncertainty in wildfire spread to the choice of *diffusion models* for segmentation without first motivating why diffusion is preferable for this task or how it performs it. Consider adding a short rationale and comparison before committing to diffusion.

- **Related work coverage (discrete diffusion):** the related-work section contrasts Gaussian vs. categorical diffusion but largely cites application-focused works; it omits key *foundational* papers on discrete/ratio/masked diffusion [1-3].

- **Learning paradigm clarity (supervised):** the method requires paired data $(y, c)$ (segmentation mask and conditioning) and is therefore supervised, but this is not stated prominently in the Introduction/Problem setup. I suggest making this explicit early (end of Intro and start of Method) and clarifying the labeling requirements.

### Minor corrections & typos

- **Line 167:**  first quotation mark (stray/incorrect quote).

### References

[1] Lou, Aaron, Chenlin Meng, and Stefano Ermon. "Discrete Diffusion Modeling by Estimating the Ratios of the Data Distribution." International Conference on Machine Learning. PMLR, 2024.

[2] Austin, Jacob, et al. "Structured denoising diffusion models in discrete state-spaces." Advances in neural information processing systems 34 (2021): 17981-17993.

[3] Shi, Jiaxin, et al. "Simplified and generalized masked diffusion for discrete data." Advances in neural information processing systems 37 (2024): 103131-103167.

**Justification:**

The paper provides a valuable empirical contribution through the introduction of **MMFire**, a controlled dataset for evaluating uncertainty in wildfire spread prediction, and a systematic study of training-free methods to enhance diversity in diffusion-based segmentation. In addition to adapting existing techniques, the authors propose a pruning method based on k-means clustering, which, while not technically novel, is well implemented and fits the paper’s purpose. The work is clearly written and reproducible, though the motivation for using diffusion models, the coverage of foundational literature, and the clarification of the supervised setup could be improved. Overall, it is a sound and practically relevant study with modest conceptual depth but strong empirical value.

---

> ### Author Rebuttal · Authors · 2025-10-22
>
> Thank you for taking the time to review our paper. We are glad that you find it valuable. Below, we explain how we will address the weaknesses you mentioned.
>
> **1. Motivating the use of diffusion models**
>
> We will expand the sentence in lines 35-36 to a short paragraph that explains that generative models can be used to generate several different outputs for a given input, and that research in recent years has demonstrated that diffusion models are particularly strong when it comes to faithfully generating images according to a given distribution.
>
> **2. Citing foundational work on discrete diffusion**
>
> Thank you for pointing out this oversight on our part. Following the feedback by you and reviewer aXxE, we will add citations to both the foundational works on Gaussian and discrete diffusion models.
>
> **3. Learning paradigm clarity**
>
> In the introduction, we will shortly mention that the experimental setup follows the supervised learning paradigm.

---

### Official Review · Reviewer_YUjj · 2025-10-10
**New synthetic benchmark dataset for multi-modal wildfire spread**

**Rating:** 2
**Confidence:** 4

**Summary:**

The paper introduces MMFire, a new synthetic benchmark dataset for evaluating multi-modal wildfire spread prediction, and comprehensively evaluates three methods for encouraging diversity in diffusion model sampling: clustering-based sample pruning, Particle Guidance, and SPELL. The core challenge addressed is generating a diverse set of predictions capturing multiple plausible future scenarios without requiring hundreds of samples. The authors demonstrate that SPELL achieves the best overall performance, improving HM IoU* by 7.5% on MMFire and 16.1% on Cityscapes compared to naive sampling, while maintaining high image quality. Experiments are conducted across three domains: wildfire prediction (MMFire), semantic segmentation (Cityscapes), and medical imaging (LIDC).

**Strengths:**

**1. Comprehensive Benchmarking Approach**

The paper makes a valuable contribution by systematically evaluating diversity-encouraging methods across three distinct domains with different characteristics. The inclusion of MMFire (8 modes, highly skewed), Cityscapes (16 modes, moderately skewed), and LIDC (4 modes, relatively uniform) allows for meaningful assessment of method robustness across different distribution types.

**2. Novel Dataset with Controlled Complexity**

MMFire fills a gap in wildfire prediction research by providing a dataset with known ground truth for all modes. The deliberate introduction of a highly skewed probability distribution creates a challenging benchmark that exposes the limitations of naive sampling strategies, where the expected number of samples to observe each mode is 307.

**3. Strong Empirical Results**

SPELL demonstrates consistent performance advantages across datasets, achieving the best balance between diversity and image quality. Notably, on Cityscapes, SPELL maintains 93.6% image quality while generating 7.3 distinct modes, compared to Particle Guidance's 82.5% quality with 15.3 modes—highlighting SPELL's superior trade-off management.

**4. Practical Insights on Method Efficiency**

The finding that diversity-encouraging methods are most effective when applied only in the first 1-2 sampling steps (Table 5, smin=40) provides actionable guidance for practitioners and significantly reduces computational overhead compared to applying this guidance throughout the entire sampling process.

**5. Well-Motivated Hyperparameter Selection**

The derivation of shield radius r₀ in Equation 9 is elegant and practical. Computing the mean minimum L2 distance among unique targets provides an interpretable, dataset-specific starting point that proved optimal or near-optimal across all datasets (Table C.1).

**Weaknesses:**

**1. Limited Real-World Validation**

The paper's most significant limitation is the lack of validation on real wildfire data. As the authors acknowledge (lines 641-643), MMFire "lacks realistic diversity." The dataset uses only 8 discrete wind directions across uniform 64×64 grids, which oversimplifies real wildfire dynamics involving:

The practical utility of these methods for operational wildfire prediction remains undemonstrated. The paper would be significantly strengthened by at least one experiment on historical wildfire data, even without complete ground truth. Adding a qualitative analysis using real wildfire satellite imagery, comparing predicted spread patterns with actual observations, or including expert evaluation of generated scenarios would benefit the work.

**2. Computational Cost Receives Insufficient Analysis**

While runtime is reported (Tables 2-3), the paper lacks a critical discussion of the practical trade-offs:

1. Clustering [256→16] takes 1h47m vs. 41m for naive sampling (4.3× overhead)
2. SPELL takes 40m vs. 41m for naive sampling (minimal overhead)
3. Is 15.5% HM IoU* improvement worth 2.6× longer runtime?

The paper positions clustering as useful when "runtime is not a big concern" (line 546) but never quantifies what level of improvement would justify specific computational costs for real-world applications.

 **3. Incomplete Analysis of Method Failures**

The paper identifies that clustering fails on skewed distributions (lines 476-490) due to "modeling error" and "distribution asymmetry," but doesn't pursue solutions. Similarly, why does SPELL outperform Particle Guidance on Cityscapes but not MMFire? The explanation offered (lines 516-526) is speculative.

Skewed distributions: The authors note clustering fails on skewed distributions. Has density-based clustering (DBSCAN, HDBSCAN) been considered, as mentioned in Future Work (line 646)? Even preliminary results would strengthen the paper.

Specific concerns:

Table 2: Particle Guidance achieves 15.3/16 distinct modes on Cityscapes but only 82.5% image quality. What specifically causes this quality degradation?

Why does the Probabilistic UNet perform so much better on LIDC (0.573) than Cityscapes (0.292)? The training instability discussion (Appendix D.1) is relegated to supplementary material.

Including failure case analysis with visualizations showing where methods produce poor-quality or duplicate samples would strengthen the analysis.

**4. Generalization to Higher Resolution Unclear**

All experiments use small resolutions (64×64 for MMFire, 64×128 for Cityscapes, 128×128 for LIDC). Real wildfire prediction requires much higher resolution. The paper provides no analysis of:

1. How computational costs scale with resolution
2. Whether early-step guidance remains effective at higher resolution

Including at least one experiment on a higher resolution image dataset to demonstrate scalability or discussing computational complexity would strengthen the analysis.

**5. Presentation Limitations**

1. Mathematical notation: Several undefined terms (e.g., σmax, σmin introduced in Appendix B but used earlier). A notation table would improve readability.
2. Figure quality: Figure 4 has very small text on the y-axis label.
3. Cross-referencing: Limited hyperlinks between the main text and the appendix. Critical information (Probabilistic UNet training issues, detailed hyperparameters) is in the appendices without clear signposting.

**6. Baseline Fairness**

The Probabilistic UNet baseline used an unofficial reimplementation with acknowledged training instability (Appendix D.1). The authors made several modifications (removed custom weight initialization, changed learning rate scheduler) that deviate from the original paper (Kohl et. Al., 2018). This raises questions about whether the comparison is fair.

Specific issue: The authors state "we generally tried to see how well the probabilistic UNet works 'out of the box'" (line 1213), but then made substantial modifications. Doubling model capacity didn't improve results, but were other architectural modifications attempted?

It is recommended to either use the official implementation recommended by the original authors in (Kohl et. Al., 2018) (https://github.com/SimonKohl/probabilistic_unet) or more clearly justify why the chosen implementation and modifications provide a fair comparison.

**7 Hyperparameter Sensitivity**

Table D.1 shows guidance strength α=10 is optimal, but only {1, 10, 100, 1000} were tested. Was a finer-grained grid search for α conducted? How sensitive are results to this choice?

**Justification:**

This paper makes solid contributions to multi-modal prediction with diffusion models and introduces a benchmark dataset. The systematic evaluation across three domains and the practical insights about early-step guidance are valuable. SPELL's strong empirical performance and its elegant, interpretable hyperparameter selection make it a promising method.

However, the paper currently falls short of publication standards for a deep learning conference venue due to:

1. Lack of real-world validation for the primary application (wildfire prediction)
2. insufficient analysis of computational trade-offs and method failure modes
3. Limited discussion of practical utility and operational considerations

To meet publication standards, the authors should:

Essential revisions:

1. Add qualitative validation on real wildfire data or expert evaluation
2. Provide a comprehensive computational cost-benefit analysis
3. Include failure case analysis with visualizations
4. Strengthen baseline comparisons (use official Probabilistic UNet implementation) (Kohl et. Al., 2018) (https://github.com/SimonKohl/probabilistic_unet).

Strongly recommended:

1. Experiment at higher resolution on at least one dataset
2. Add mode coverage and mode-weighted metrics
3. Expand discussion of when multi-modal prediction matters operationally

Minor improvements:

1. Improve table/figure formatting and cross-referencing
2. Add notation glossary
3. Move critical implementation details from the appendix to the main text

With these revisions, this would be a strong contribution demonstrating both methodological advances and practical applicability. The current version presents good ideas that need more thorough validation and analysis.

---

> ### Author Rebuttal · Authors · 2025-10-22
>
> Thank you for investing so much of your time into giving us feedback on our paper. We felt that all of the feedback was sensible, even if we will push back on some of it below. For us, this paper is a stepping stone on the path to multi-modal wildfire modeling with diffusion models. It establishes how we can eventually use the investigated methods on real-world data, where we will not have the multiple ground truth futures to assess whether the diverse outputs are correct or not. Similar to reviewer aXxE, you want to see whether these methods work on real data. To us, this is exactly the direction we want to pursue, but we believe that this requires a sizable amount of further work that would go beyond the limits of this paper.
>
>
> **1. Add qualitative validation on real wildfire data or expert evaluation**
>
> To properly evaluate the methods we investigated in this paper on real wildfire data, we would first require a diffusion model trained on this data, that accurately reflects the learned distribution. The closest model that exists for this is the one proposed by Yu et al. [1], but as of this writing, this paper is only a preprint and the corresponding code and model do not seem to be publicly available. Furthermore, the model was only trained on simulated data, and thus does not represent real wildfire data. We are planning to dedicate a future paper to the problem of training and evaluating such a model and will then of course apply the knowledge we have gained in the current paper (as mentioned in ll. 74f.).
>
>
> **2. Provide a comprehensive computational cost-benefit analysis**
>
> Which trade-offs between computation, number of samples, and performance are acceptable completely depends on the specifics of the application in question. This makes it very difficult to conduct an analysis in the paper that actually provides useful insights. In autonomous driving, the clustering approach will likely never be chosen, since real-time results are important. In wildfire spread, predictions are performed on the order of tens of minutes to several hours in advance. Thus the additional time needed for clustering would likely be acceptable. How the different factors should be traded off is also impacted by factors like the number and types of available GPUs (sampling can be parallelized), the required time to capture and load input data from cameras or satellites, as well as permissible delay to respond to the input. Such cost-benefit analyses thus seems too application-specific to conduct in a meaningful way in the context of this paper.
>
> **3. The paper identifies that clustering fails on skewed distributions (lines 476-490) due to "modeling error" and "distribution asymmetry," but doesn't pursue solutions.**
>
> The underlying problems of models being imperfect and the distribution being asymmetric are part of the conditions that we assume as fixed, due to the given data distribution and trained diffusion model. In this paper we found how well KMeans with Chamfer distance works and where its problems are. There are various ways that might improve upon this, same as for the other methods that do not deliver perfect results. For this paper, we felt that the thorough comparison of the presented methods should be the focus and thus decided that the various possible improvements of the investigated methods were out of scope.
>
>
> **4. Table 2: Particle Guidance achieves 15.3/16 distinct modes on Cityscapes but only 82.5% image quality. What specifically causes this quality degradation?**
>
> HM IoU* measures a combination of image quality and diversity. If you imagine the Pareto front showing the trade-off between sample quality and diversity, this result from Table 2 would likely be on the extreme end of high diversity, low quality, while most other results seem to stay within areas of higher quality. The general cause for this is that particle guidance simply repels images from each other, without any regard for the resulting image quality. We can still assure some amount of quality by tuning the alpha parameter to not push too hard, and by allowing the learned score function to remedy the damage to image quality caused by moving samples into somewhat random directions. For most of our experiments, image quality happened to remain rather high. But given how particle guidance works and that we perform hyperparameter selection with regards to HM IoU*, which combines quality and diversity, it is plausible that sometimes points on the Pareto curve can be reached that favor diversity over quality.
>
> **5. Add mode coverage and mode-weighted metrics**
>
> We report the number of distinct modes, which is a direct measure of mode coverage. HM IoU* is a measure of how well each equally-weighted mode is represented in the generated samples. Unlike the number of distinct modes, it not only considers which mode a sample is closest to, but how well it represents that mode.
>
> **6. Include failure case analysis with visualizations**
>
> With the extra page of space upon acceptance, we will visualize examples in the main paper, comparing a batch of samples generated with naive sampling with one generated with SPELL, starting from the same latent noise. Comparisons with other methods will likely be added to the appendix.
>
> **7. Strengthen baseline comparisons (use official Probabilistic UNet implementation)**
>
> We opted for the inofficial Pytorch implementation of the Probabilistic UNet, because all of our code is based on Pytorch and Pytorch Lightning. Switching to Tensorflow, which the official implementation uses, would add more sources for possible errors, which we wanted to avoid. Furthermore, the results we received on LIDC were very similar to the results of the Probabilistic UNet paper, which we took as an indication that the implementation is correct. However, for datasets that were not LIDC, the training had proven very unstable, often resulting in NaN losses, which made us introduce several modifications that you criticized.
>
> After receiving your comments, we re-examined the different code bases and found small misalignments between them. Most importantly, we found that a logarithm operation was responsible for the training instability we experienced that had made the various changes necessary in the first place. Having solved this instability by adding a small epsilon term, we have now been able to increase the single-batch performance on Cityscapes from 0.292 to 0.345 HM IoU* in a trial run. There remains a gap to the 0.416 HM IoU* achieved by naive sampling from the diffusion model, but given that the segmentation masks in our datasets are much more diverse and the distribution across masks much more asymmetric than those of LIDC, it does not seem unlikely to us that the probabilistic UNet is simply not as capable of capturing this difficult distribution as the more recent diffusion models.
>
> We will update the results for Cityscapes and MMFire accordingly when the full results are available. Finding this error and completely retraining and evaluating the model was not possible within the rebuttal period.
>
> For the re-run of the probabilistic UNet experiments, we use the same architecture and weight initialization as the original implementation. However, the learning rate schedule for LIDC simply decreases the learning rate at fixed step numbers. These fixed points would likely not be optimal for the other datasets. We therefore keep using OneCycleLR for the non-LIDC datasets instead, which does not require us to choose such arbitrary points for these other datasets.
>
> **8. Generalization to Higher Resolution**
>
> When moving to higher resolutions, the main additional cost is incurred by the diffusion model. This cost of course affects the clustering method the most, since it requires denoising a large number of samples at the initial step. To a lesser degree it also affects particle guidance, which takes a backpropagation step through the diffusion model to compute the guidance term. Furthermore, all methods compute pairwise image distances, which uses computation according to O(b²d), if b is the number of samples in a batch that are compared, and d is the number of pixels. However, these characteristics mostly refer to the original methods, and thus should be addressed in their respective original publications. We will add a short paragraph in the appendix explaining the above and mention that our clustering method would be most strongly impacted by the increase in computation cost incurred by higher resolutions.
>
> **8.1 All experiments use small resolutions (64×64 for MMFire, 64×128 for Cityscapes, 128×128 for LIDC). Real wildfire prediction requires much higher resolution.**
>
> The necessary pixel resolution depends on the spatial resolution of the sensors used. Among the respective studies we referenced, the highest resolution is achieved by the VIIRS satellite used in WildfireSpreadTS, which has a resolution of 375m x 375m per pixel. A 64x64 pixel image thus would have a side length of 24km, which appears adequate. At higher resolutions, it would also be possible to crop images to the resolutions used in our paper, given that the fire fronts at opposite edges of the image will likely have little influence on each other, due to the large geographical distance.
>
> **9. Presentation Limitations**
>
> Thank you for pointing these limitations out, they will be addressed.
>
> **10. Hyperparameter Sensitivity of Particle Guidance**
>
> We will expand upon the results in Table D.1 with a more fine-grained hyperparameter grid search for alpha in [1,100].
>
>
> [1] W. Yu, A. Ghosh, T. S. Finn, R. Arcucci, M. Bocquet, and S. Cheng, “A Probabilistic Approach to Wildfire Spread Prediction Using a Denoising Diffusion Surrogate Model,” Jul. 01, 2025, arXiv: arXiv:2507.00761. doi: 10.48550/arXiv.2507.00761.

---

### Official Review · Reviewer_aXxE · 2025-10-10

**Rating:** 1
**Confidence:** 4

**Summary:**

This paper investigates training-free methods to increase sample diversity in conditional diffusion models, with the stated goal of modeling uncertain wildfire spread scenarios. The authors introduce a new simulated dataset, MMFire, for this purpose. They empirically evaluate three sampling methods—Particle Guidance (PG), SPELL, and a novel clustering-based pruning approach—against a naive sampling baseline. The evaluation is conducted on MMFire, the LIDC medical dataset, and a synthetically modified version of Cityscapes to assess cross-domain transferability. The paper concludes that diversity-enhancing methods, particularly SPELL, significantly improve the trade-off between sample diversity and quality compared to the baseline.

**Strengths:**

- Addresses a Critical Application: The paper tackles the important and challenging problem of uncertainty quantification in wildfire modeling. The goal of generating a diverse set of plausible scenarios is a well-motivated and practical objective for creating effective decision-support tools.

- Contribution of a New Dataset: The creation and proposed release of the MMFire dataset is a contribution. A public, standardized benchmark is essential for advancing research in this specific domain, and MMFire can serve as a useful starting point for future studies.

- Systematic Empirical Study: The paper provides a helpful comparison of several training-free diversity-enhancement techniques. By evaluating these methods across three distinct datasets, the authors offer insights into their relative performance, costs, and trade-offs, which can inform future work.

**Weaknesses:**

- Fundamental Disconnect Between Motivation and Evaluation: The central motivation is wildfire prediction, a complex physical forecasting problem. However, the evaluation framework is diluted with tasks that are fundamentally different in nature. The LIDC task concerns inter-observer ambiguity in static medical images, while the Cityscapes task—generating masks by randomly toggling object classes—is a highly contrived problem that bears no resemblance to the temporal evolution of a wildfire. This mismatch undermines the paper's core claim that its findings are directly relevant to the wildfire domain.


- Limited Realism of the Primary Dataset: The authors explicitly state that the MMFire dataset is "not meant to be perfectly realistic". Uncertainty is injected by varying a single, global wind direction parameter, which is a significant oversimplification of the complex, localized, and multi-faceted sources of uncertainty in real-world fire dynamics. The validity of conclusions drawn from this simplified "toy problem" and their transferability to the real world is therefore questionable.



- Lack of Methodological Clarity and Novelty: The paper's methodological presentation is imprecise and its novel contribution is weak.

1) The discussion around guidance is potentially misleading. Equation (4) presents a generic formulation for guided diffusion. However, the guidance term g(x; t) in equation 4 is not derived from a conditional score (as in classifier or classifier-free guidance).  The paper fails to clearly articulate this, potentially confusing readers who are familiar with traditional guidance mechanisms.

2) The main novel method proposed by the authors, a clustering-based pruning approach, is a straightforward heuristic. More importantly, their own analysis reveals its primary shortcoming, stating that the "clustering algorithm used is not able to reliably detect the modes of the distribution" . A proposed method that fails at its central objective is a significant weakness.


- Insufficient Rigor and Precision: The manuscript suffers from a lack of formal precision and rigorous justification expected in a scientific publication.

1) The paper begins with informal, speculative language, such as "We believe that this might be related to...", where more scientific phrasing like "We hypothesize..." would be appropriate.

2) Key technical concepts are introduced abruptly and without proper initial context or citation. For instance, the term "diffusion model" is used in the abstract and introduction without immediate citation to a foundational work. The paper also jumps from the high-level problem of "wildfire spread" to the specific technical formulation of "segmentation masks" without sufficiently motivating why this is the chosen representation of a fire's outcome.


3) Key design choices are justified anecdotally by referencing "preliminary experiments" without presenting any supporting data. This lack of transparency in dismissing alternative methods or selecting specific parameters makes it impossible for the reader to verify the claims and weakens the paper's overall credibility.

**Justification:**

The paper is premised on addressing the critical challenge of wildfire prediction, but this premise is not supported by the experimental design. The evaluation relies on a combination of an admittedly unrealistic primary dataset and two unrelated tasks, which invalidates the claims regarding the method's utility for the target application. Furthermore, the paper's novel methodological contribution is shown to be ineffective by the authors' own admission.

---

> ### Author Rebuttal · Authors · 2025-10-22
>
> Thank you for taking the time to provide us with such detailed feedback on our research. Your main criticism seems to be that we motivate our approach with the application to wildfire spread, but that we then do not end up solving the problem of multi-modal wildfire spread on realistic data. This disconnect that you see is a valid perspective. We will try below to convince you that an alternative perspective focusing on the method is also valid. From this alternative perspective, we believe that our paper provides enough value to warrant publication.
>
> **1. Limited Realism of the Primary Dataset and Disconnect Between Motivation and Evaluation**
>
> The problem we are facing when trying to approach multi-modal wildfire modeling is that no adequate dataset exists to conduct such research on. The real world observational data only provides a single future for each starting condition. How to obtain a dataset that is both realistic and multi-modal is an open question, as far as we are aware.
> Our alternative approach is thus to assume that, trained on enough single-future observational data, a generative model might be able to generate diverse outputs that are aligned with the ground truth possible futures (ll. 71ff.). However, without knowing these possible futures, we are not able to evaluate whether the training of and sampling from the model lead to the generation of plausible or implausible futures. Existing literature on generating segmentation masks mostly ignores the problem of ensuring adequate diversity (ll. 37ff.). Thus, we create a dataset (MMFire), to evaluate how we can reach this sample diversity. MMFire is not both multi-modal and realistic, because it is unclear how to  achieve this. Instead, we sacrifice some realism in the mode diversity to ensure that we can draw methodological conclusions from our experiments that are likely to transfer. To increase the evidence for transferability, we use three very different datasets for our experiments.
>
> From a perspective of _data generation_, you are completely correct that the sources of uncertainty between the different datasets are not aligned with realistic sources of uncertainty in wildfire spread. However, from a _methodological perspective_, the datasets all represent ambiguous segmentation tasks, independent of what the original source of ambiguity was. By evaluating the methods, and the specific way that we use them for segmentation masks, on a range of ambiguous segmentation tasks, we can draw methodological conclusions that are more likely to transfer.
>
> Ultimately, our goal with this paper is to gain methodological knowledge that will _eventually_ be useful for wildfire spread prediction. We thus orient many choices to align with this goal. But due to the inherent difficulty of this long-term goal, our paper only represents a stepping stone on this path.
>
> **2. Novel contribution is weak.**
>
> We agree that the proposed clustering-based pruning approach is a very straightforward approach. While it does not greatly outperform the other methods, it has the unique advantage over the other methods that it leaves the original sampling trajectory unmodified. Since such modifications always involve the risk of impacting image quality, this can be a valuable property for applications that are very sensitive to reduced image quality.
>
> Our phrase "clustering algorithm used is not able to reliably detect the modes of the distribution" will be rephrased, given how you, as the reader, seem to have understood it. It was only supposed to communicate that the clustering algorithm does not perfectly separate all of the available modes into separate clusters, as indicated by the gap in HM IoU* between cluster centers and the full set of samples. It is not unusual for a clustering algorithm to imperfectly separate real data. Reducing the data to cluster centers will usually lead to a reduction in performance quality, given that this reduction discards a lot of information. However, this performance gap does not imply that the clustering approach is not useful. In fact, our approach clearly outperforms naive sampling in all experiments, thereby achieving its purpose of increasing diversity within a fixed number of samples.
>
> Of course, as we state in the paper: "on MMFire, [the clustering-based approach] is clearly outperformed by both particle guidance and SPELL." (ll.517f) The approach does not represent a new state of the art. However, this approach is only one of multiple contributions, not the focus of the paper. The main focus is the transfer of sample-diversity-enhancing methods to the domain of segmentation masks, and identifying changes necessary to make these methods work well on this domain, motivated by the application to wildfire spread modeling. As long as the clustering-based approach adds _some_ value, we believe it should not be taken as an argument against publication of the paper.
>
> **3. Missing citations for foundational work**
>
> Thank you for pointing out this oversight on our part. Following the feedback by you and reviewer N3hm, we will add citations to both the foundational works on Gaussian and discrete diffusion models.
>
> **4. References to preliminary experiments**
>
> We referred to preliminary experiments on CADS and on choosing a distance metric for our clustering approach. The main motivation for the short references to preliminary experiments was to include the empirical knowledge we had gained for the benefit of other practitioners, without overloading the paper with information that has little influence on the methods in the focus of the paper. The difference between L2 and Chamfer distance was rather small, and CADS very clearly did not work and was thus not worth pursuing. We will add respective experimental results to the appendix for completeness.

---

### Meta-Review · Area_Chair_qde7 · 2025-10-31

**Recommendation:** Accept (Poster)
**Confidence:** 4

**Metareview:**

This paper presents an empirical study on training-free diversity enhancement for diffusion models, motivated by wildfire prediction. Reviewers were split, but mostly positive, praising the systematic benchmarking, the new MMFire dataset, and the practical insights. The primary concern was a disconnect between the wildfire motivation and the simplified, multi-domain experimental setup. The authors provided a convincing rebuttal, successfully reframing the work as a crucial methodological stepping stone for the broader challenge of ambiguous segmentation, a necessary precursor to tackling real-world applications. Having addressed this and other key points, the view of the AC shifted in favour of the paper's methodological contributions. While many of the concerns raised in the reviews are valid, the AC agrees with the support acceptance, and encourages the authors to implemented the promised changes tto improve the manuscript.

---

### Decision · Program_Chairs · 2025-11-05

**Decision:**

Accept (Poster)

**Comment:**

We recommend a poster presentation given the AC and reviewers recommendations.